**High temporal resolution hydrometeorological data collected in the tropical**
**Cordillera Blanca, Peru (2004-2020)**
Emilio I. Mateo[1], Bryan G. Mark[1], Robert Å. Hellström[2], Michel Baraer[3], Jeffrey M. McKenzie[4],
Thomas Condom[5], Alejo Cochachín Rapre[6], Gilber Gonzales[6], Joe Quijano Gómez[6], Rolando
Cesai Crúz Encarnación[6]
[1]Department of Geography, Byrd Polar and Climate Research Center, The Ohio State University,
Columbus, OH, USA
[2]Department of Geography, Bridgewater State University, Bridgewater, MA, USA
[3]Département de génie de la construction, École de technologie supérieure, Montreal, QC,
Canada
[4]Department of Earth and Planetary Sciences, McGill University, Montreal, QC, Canada
[5]Université Grenoble Alpes, CNRS, IRD, Grenoble-INP, Institut des Géosciences de
l'Environnement (IGE, UMR 5001), Grenoble, France
[6]Peruvian National Water Authority, Division of Glaciers and Water Resources, Huaraz, Peru
*Correspondence to*: Emilio Mateo (mateo.9@osu.edu)
**Abstract.** This article presents a comprehensive hydrometeorological dataset collected over the
past two decades throughout the Cordillera Blanca, Peru. The data recording sites, located in the
upper portion of the Rio Santa valley, also known as the Callejon de Huaylas, span an elevation
range of 3738 - 4750 m a.s.l. As many historical hydrological stations measuring daily discharge
across the region became defunct after their installation in the 1950s, there was a need for new
stations to be installed and an opportunity to increase the temporal resolution of the streamflow
observations. Through inter-institutional collaboration the hydrometeorological network
described in this paper was deployed with goals to evaluate how progressive glacier mass loss
was impacting stream hydrology, and to better understand the local manifestation of climate
change over diurnal to seasonal and interannual time scales. The four automatic weather stations
supply detailed meteorological observations, and are situated in a variety of mountain
landscapes, with one on a high-mountain pass, another next to a glacial lake, and two in glacially
carved valleys. Four additional temperature and relative humidity loggers complement the
weather stations within the Llanganuco valley by providing these data across an elevation
gradient. The six streamflow gauges are located in tributaries to the Rio Santa and collect high
temporal resolution runoff data. The datasets presented here are available freely from
https://doi.org/10.4211/hs.35a670e6c5824ff89b3b74fe45ca90e0 (Mateo et al., 2021). Combined,
the hydrological and meteorological data collected throughout the Cordillera Blanca enable
detailed research of atmospheric and hydrological processes in tropical high-mountain terrain.












## 1 Introduction

Glaciers and water resources in the Cordillera Blanca, Peru, have been under close observation for nearly a century. In the 1930s, an Austrian geographer from Universitat Innsbruck, Hans Kinzl, laid the groundwork for glaciological research in the region by surveying and mapping the glaciers and identifying other natural features in the mountainous landscape (Kaser and Osmaston, 2002). While the first systematic Peruvian effort to observe glacier tongue variations in the Cordillera Blanca was initiated in 1944 by Broggi (Petersen et al., 1969), glaciers as a source of security and water resources became the object of study in the subsequent decades. The Unidad de Glaciología e Hidrología (later Unidad de Glaciología y Recursos Hidricos (UGRH) of Electroperú S.A.) was initiated by the Corporación Peruana del Santa, a government company for energy development, and took on glacier and lake monitoring after many glacial lake outburst flooding (GLOF) events damaged communities downstream (Ames, 1998). GLOFs in 1941, 1945, and 1950 killed over 6000 people and destroyed a third of the Ancash district capital of Huaraz (Carey, 2010; Carey et al., 2012). Other glacier hazards have had detrimental impacts on the communities as well, including the 1970 avalanche, triggered by a massive earthquake, which killed approximately 6000 people and along with the debris flow it produced, covered the city of Yungay (Carey, 2010). Lliboutry, Morales and Schneider (1977) investigated two glaciers in the mountain range in relation to the danger they presented for flooding a downstream power plant. From the 1970s to the 1990s, a local Peruvian from Huaraz, Alcides Ames, provided many important contributions to the present knowledge of glaciers in the Cordillera Blanca while employed at UGRH, and after his retirement continued to dedicate much of his time to studying the glaciers and sharing his knowledge with other researchers (Ames et al., 1989; Hastenrath and Ames, 1995; Ames and Hastenrath, 1996; Ames, 1998; Kaser and Osmaston, 2002). In the 1980s, Georg Kaser, an Austrian geographer, began studying the Cordillera Blanca with a focus on the extent, causes, and possible consequences of the observed glacier retreat (Kaser and Osmaston, 2002). For many decades, much of the research in the region focused on developing better maps of the mountains and observing the marked glacier retreat in the tropics. Although the meteorological and hydrological stations installed by UGRH had been in place for nearly 40 years, it was not until the mid-1990s when studies began to assess water resources in the Cordillera Blanca, but with minimal focus on flooding and GLOFs (Kaser and Georges 1997; Mark and Seltzer, 2003).

The UGRH spent many years collecting an abundance of climatological, hydrological, and glaciological data, which was useful for future researchers, hydroelectric and mining companies, and other stakeholders throughout the Cordillera Blanca and its glaciers. Since the mid-1990s, most of the daily-resolution discharge data from the Rio Santa watershed has been cataloged and collected at a much-reduced number of sites by private energy companies who took over after Electroperu was privatized. The rapid turnover of the energy companies holding the discharge observations makes it difficult to track down all available historical data. Also, many of the hydrological stations have become defunct since they were installed in the 1950s,

creating a need for new stations to be installed for continuous monitoring of the region. A further
benefit of installing a new hydrological monitoring network in the 2000s was to significantly
increase the temporal resolution of the discharge observations.

Collaborating researchers from The Ohio State University, Bridgewater State University

and McGill University initiated a network of embedded environmental sensors in 2005. The
main goals of the instrument arrays were to evaluate how progressive glacier mass loss was
impacting stream hydrology, and to better understand the local manifestation of climate change
on the variability and controls of local weather phenomena over diurnal to seasonal and
interannual time scales. In order to create a sustainable network for continuous, long-term
observation, this project has been maintained in close collaboration with Peruvian researchers
and government agencies, as well as with other international scientists to leverage resources in
maintaining instruments, in exchange for openly sharing data.

The instrumentation for the data collection we present in this work was installed and

maintained under a collaborative work agreement ("convenio") formalized with the Peruvian
government agency overseeing the office in Huaraz (Peruvian Institute of Natural Resources
(INRENA) and the Autoridad Nacional del Agua (ANA)). This work agreement involved a
secondary collaboration with other international researchers who shared in installing and
calibrating the instrumentation. Specifically, Dr. Thomas Condom of the French Institut de
Recherche pour le Developpement (IRD) joined the agreement to install and maintain a series of
stream gauges logging water stage at 15-minute intervals. We also collaborated with the Austrian
research team of Dr. Irme Juen and Professor Georg Kaser from the Universitat Innsbruck,
Austria, who co-located a precipitation gauge with our weather station in Llanganuco. Further
details about the instrumentation are provided below.

Inter-institutional collaboration in this fashion has provided an effective partnership to

aid in maintaining the instrumentation over time, but also introduces challenges of logistical
coordination and data continuity. Visitation as non-resident international scientists to the field
sites has been feasible only one or two times annually. Thus, the Peruvians in our collaboration
have incorporated our instruments within their routine monitoring network. This has permitted
regular observations of stations and instruments to download data loggers, perform stream
discharge measurement to build a rating curve, and undertake a limited range of repair work.
However, limitations in local resources and manpower in Peru have often prevented recordings
of stage observation and discharge measurement to constrain rating curves. Likewise, having
multiple operators also increases some risk for data recovery errors. For instance, loggers that are
improperly relaunched after data downloads can jeopardize subsequent acquisitions. Having
more frequent site visits can allow for interventions, but also incurs increased risk of operator
error. Furthermore, Peruvian domestic political changes have disrupted the operations by
introducing different leadership, with altered operational priorities and resources, which directly
or indirectly interrupt the continuity of trained personnel responsible for data recovery and
preservation. Overcoming the many challenges of maintaining the instrumentation and
constructing rating curves has required regular cross-cultural communication, multiple and
annual visitation to the region. Likewise, the rating curves continue to be refined as additional
measurements are made.

Our collaborative observations have provided important new insights into how the
hydroclimate of the region is changing on different scales. Updated discharge and climate
observations in specific glacier catchments documented important shifts in seasonal supply of
glacier storage to the Yanamarey catchment (Bury et al., 2011) as well as suggesting regional
thresholds in glacier melt provision (Mark et al., 2010). The discharge constraints also provided
important validation for a novel hydrochemical basin characterization method to quantify
proportionate glacier melt and groundwater contributions to streamflow in tributaries of varying
glacierized coverage (Baraer et al., 2009, 2015). Regionally, the gauge network provided the key
constraint for a model of time-progressive hydrograph evaluation that verified significantly that
the main catchment had already passed "peak water" in the wake of strong glacier recession
underway for multiple decades (Baraer et al., 2012). Our embedded temperature and humidity
loggers distributed over elevation and linked to weather stations in the Llanganuco valley have
revealed novel diurnal to seasonal variations in lapse rates linked to catchment-specific valley
wind dynamics validated with downscaled climate models (Hellstrom et al., 2017). These studies
demonstrate the importance of collecting in situ hydrometeorological data and indicate the need
for continued data collection in the high Andes (Condom et al., 2020).

In this paper, we document available data from the Cordillera Blanca area collected over
the past 15 years. It is separated into (i) meteorological data recorded by permanently installed
automatic temperature and relative humidity loggers (Lascars), or automatic weather stations,
and (ii) hydrological data consisting of stage and discharge data from multiple sub-catchments.
The data are stored in .csv extension format on Consortium of Universities for the Advancement
of Hydrologic Science, Inc. (CUAHSI) Hydroshare:
https://doi.org/10.4211/hs.35a670e6c5824ff89b3b74fe45ca90e0 (Mateo et al., 2021). The data
collection consists of multiple time series of point observations from both meteorological and
hydrological measurement sites. These high-temporal resolution data provide detailed insight
into the complex hydro-meteorological system of the tropical Cordillera Blanca.
**2 Study Area - Cordillera Blanca**

The Rio Santa (Santa River) captures runoff from the western side of the glacierized
Cordillera Blanca and the eastern side of the non-glacierized Cordillera Negra, encompassing a
drainage area of 11636 $km^2$ at the outlet to the Pacific Ocean. While the headwaters of the Rio
Santa are found at Laguna Conococha at 4100 m above sea level (a.s.l.), the highest point in the
basin (and in Peru) is the summit of Huascaran at 6768 m a.s.l. The average slope of the entire
drainage basin is 20.6° and its average elevation is 3374 m a.s.l. calculated from a 3 m resolution
DEM of the region. The Rio Santa flows over 300 km northwest from its origin at Laguna
Conococha, an alpine lake at 4000 m a.s.l., to its outlet into the Pacific Ocean, near Chimbote.
The Callejon de Huaylas refers to the upper section of the Rio Santa, comprising approximately
4773 $km^2$ located above the Cañon del Pato 50 MW hydroelectric generation plant in Huallanca
(Figure 1). The Rio Santa basin is home to three other hydroelectric plants and provides water to
the expansive Chavimochic irrigation district near the coast (Mark and McKenzie, 2007). The
discharge on the Rio Santa has been carefully monitored since the Cañon del Pato dam began
operating in the 1950s, however, only one station near the dam remains active which is situated
slightly upstream at La Balsa (labeled as LBQ in Figure 1).
Figure 1.
Map of the Callejon de Huaylas showing the locations of the hydrological and meteorological
stations. (Base layers of the map originated from: © Esri, © NASA, © NGA, © USGS, © FAO,
© NOAA; all other layers were created and edited by authors of this article.)

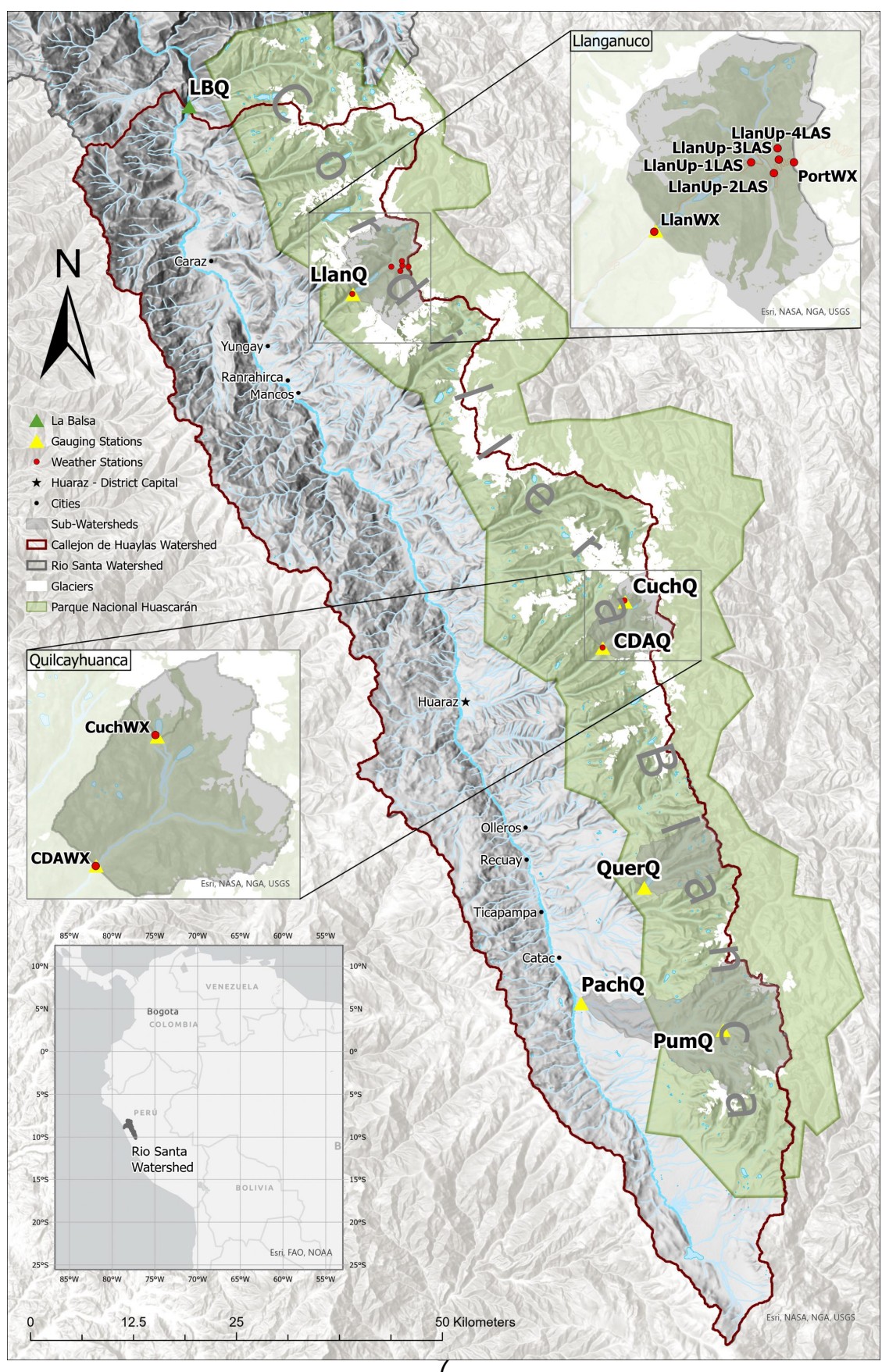


Rio Santa discharge experiences a strong seasonal contrast with the lowest discharge
occurring between July and September, while peak discharge, nearly 20 times greater, typically
occurs in March. Calculated from daily streamflow observations between 1954 and 2015, mean
annual streamflow at La Balsa station on the Rio Santa, is 87 $m^3s^{-1}$, while the average annual
minimum discharge is 25 $m^3s^{-1}$, and the average annual maximum discharge is 445 $m^3s^{-1}$. The
Callejon de Huaylas is approximately 10% covered in ice (RGI Consortium, 2017), has an
average slope of 19° and an average elevation of 4055 m a.s.l. Glacial melt in the Rio Santa at La
Balsa provides 10-20% of the total annual discharge and may exceed 40% in the dry season
(Mark et al., 2005; Condom et al., 2012).

The climate of the Callejon de Huaylas is semi-arid and displays distinct precipitation
seasonality, with the wet season between October and May being responsible for 80% of the
800-1200 mm/year of precipitation (Baraer et al., 2009), and the dry season lasting from June to
September (Figure 2). As a localized example, the Llanganuco valley displays an average total of
8 mm during the dry season, and an average of total of 258 mm during the wet season months of
December, January, and February, based on monthly totals from 1953 to 2010 (Hellstrom et al.,
2017). Variations in river discharge are largely driven by seasonality of precipitation.
Temperature remains nearly constant in the outer tropics, with the annual variation in
temperature being smaller than diurnal variation (Kaser et al., 1990).
Figure 2.
A climograph from LlanWX in the Cordillera Blanca displaying the strong contrast in
precipitation between wet and dry seasons, while maintaining a steady temperature throughout
the entire year.

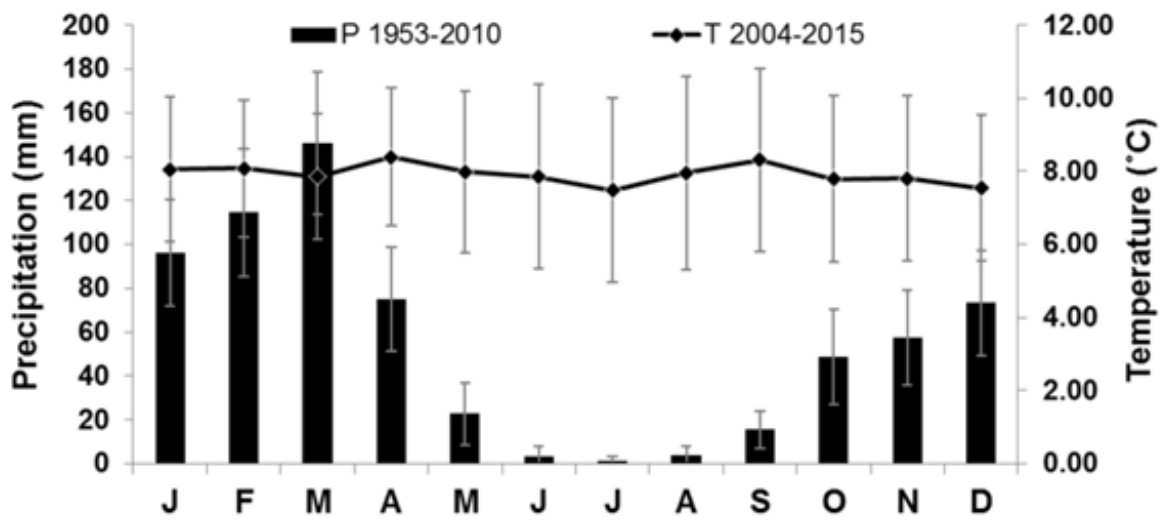


The geology in the Rio Santa basin is dictated by numerous tectonic and erosional
processes because the basin is situated along the active detachment fault of the Cordillera Blanca
(McNulty et al., 1998; Garver et al., 2005; Eddy et al., 2017). The highest peaks of the Cordillera
Blanca are composed largely of a granodiorite batholith intruded into a metamorphic unit that
includes hornfels, gneiss, and sulfide-rich lithologies, such as pyrite schist, phyllite, and pyrite-
bearing quartzite (Giovanni et al., 2010; Eddy et al., 2017). Granodiorite is found predominantly
in the northern portion of the range, while the southern portion of the range is made up of a
variety of metasediments, including quartzites and carbonates (Garver et al., 2005). The main
valley of the Rio Santa watershed is covered by recent sediment deposits, including alluvium,
landslide deposits, and glacial-fluvial fill. The impact of past and present glacier extent in the
topography is visible in geomorphic features throughout the range, including steep walled U-
shaped valleys, moraines, and proglacial lakes (Eddy et al., 2017).
**3 The Data**
In the following section, the data collected from the Cordillera Blanca are presented in
two collections. First, we provide a description of the meteorological stations and embedded
sensor network of Lascar data loggers, and then we present the time series data from the stations.
Second, we provide details of the setup and context of the discharge gauging station network,
and then conclude by presenting the time series of discharge data and general statistics from
them. The Figures A1, A2, and A3 in the Appendix, indicate missing data beyond 2020, however
due to travel restrictions over the past two years we have not been able to collect the up-to-date
measurements. Our intent is to sustain these hydrological and meteorological measurements into
the future once travel to the region is possible again. This future data will be available upon
request as it is collected.
**3.1 Meteorological Data**
The Servicio Nacional de Meteorología e Hidrología del Perú (SENAMHI) has operated
a relatively dense national network of meteorological stations since 1964. Air temperature is
provided as daily $T_{max}$ and $T_{min}$ (additionally measured at 07:00, 13:00 and 19:00 local time), and
precipitation is measured once a day. Some SENAMHI stations still provide daily temperature
and precipitation data through their web portal, providing data back to 1980
(https://www.senamhi.gob.pe/?&p=descarga-datos-hidrometeorologicos), but very few stations
within the Rio Santa Valley or Cordillera Blanca, Peru, remain active. Note that additional data
from SENAMHI are available upon request from the Swiss MeteoDat GmbH team through a
data portal (Schwarb et al. 2011, http://www.meteodat.ch/). In addition, the Universidad
Nacional Santiago Antúnez de Mayolo (UNASAM) has maintained a network of what was
originally 16 meteorological stations located at different elevations in the Cordillera Blanca
(Ancash district) since 2012. Monthly totals of precipitation data for the Rio Santa watershed
have been collected by Electroperú South America since 1953 although measurements were
interrupted in the mid-1990's with the privatization of the respective institutions when most of
the stations were abandoned and quality control was an issue (Kaser et al., 2003). More recent
installations of weather stations in the Cordillera Blanca have been mostly associated with short-
lived research projects typically lasting less than three years (Hofer et al., 2010; Georges and
Kaser, 2002). The hourly meteorological observations we describe below were collected
primarily from instrumentation we installed in two sub-catchments of the Rio Santa drainage
basin, Llanganuco, and Quilcayhuanca.

The Llanganuco valley is situated on the western side of the Cordillera Blanca across a
southwest (~240°) to northeast (~60°) axis with elevations ranging from 3400 m a.s.l. to 6746 m
a.s.l. Llanganuco is one of most glacierized valleys in the mountain range at 30% glacier
coverage, making it one of the most glacierized tropical valleys in the world. In the Llanganuco
catchment, our group has installed multiple weather stations dating back to 2007, although only
one remains active. This currently active station, labeled LlanWX, is located at 3835 m.a.s.l. near
the lower of the two largest valley lakes (Table 1).
Table 1.
This table provides general information about each meteorological station (WX) and lascar data
logger (LAS) location within the embedded sensor network. Dates with an asterisk in the "Period
of Operation" column indicate the station is operational.

| Station | Station Valley | Elevation (m a.s.l.) | Period of Operation | Lascar Error Adjustment | Slope Angle (°) | Slope Aspect (°) | Temporal Resolution |
|---|---|---|---|---|---|---|---|
| Cuchillacocha - CuchWX | Quilcayhuanca | 4642 | 2013-2020* | -- | 0 | NA | 30-min |
| Casa de Agua - CDAWX | Quilcayhuanca | 3924 | 2013-2019* | -- | 0 | NA | 30-min |
| Llanganuco - LlanWX | Ranrahirca | 3835 | 2007-2019* | Yes | 0 | NA | 60-min |
| Portachuelo - PortWX | Ranrahirca | 4750 | 2006-2015 | -- | 0 | NA | 60-min |
| LlanUp-1/1A LAS | Ranrahirca | 3955 | 2006-2015; 2015-2020* | No | 14 | 301 | 60-min |
| LlanUp-2/2A LAS | Ranrahirca | 4122 | 2006-2014; 2015-2020* | No | 33 | 259 | 60-min |
| LlanUp-3/3A LAS | Ranrahirca | 4355 | 2006-2015; 2015-2020* | No | 31 | 236 | 60-min |
| LlanUp-4/4A LAS | Ranrahirca | 4561 | 2006-2015; 2018-2020* | Yes | 33 | 225 | 60-min |


The embedded sensor network (ESN), as described by Hellström et al. (2010) and
Hellström and Mark (2006), provides the in-situ meteorological data provided in the paper. In
July 2007 the first automatic weather station (AWS), LlanWX, was installed near the lower lake
in the Llanganuco valley to collect a continuous record of point measurements of air
temperature, wind speed, wind direction, relative humidity, and solar irradiance (Table 2). The
AWS shown in Figure 3A is located in an open area on the valley floor and is surrounded by
*Polylepis* trees. A lascar data logger was also hung at this site for static calibration. The site is
protected in Huascaran National Park, and the location was previously used by the University of
Innsbruck for precipitation measurements. The most significant wind obstructions are the steep
bedrock walls of the valley toward the northwest and southeast which exceed 1000 meters above
the valley floor. Northerly and southerly winds are occasionally recorded by LlanWX and are
likely caused by turbulence or lateral winds from the uneven heating of the valley walls. Winds
flow parallel to the axis of the valley during approximately 92% of the recorded time and are not
greatly obstructed by surface vegetation. The sensors for LlanWX were originally sourced from
the Onset Computer Corporation and logged with an Onset HOBO® data logger
(http://www.onsetcomp.com) until 2014 when these loggers were replaced with an Iridium®
satellite Data Garrison logger (http://www.upwardinnovations.com). A 102-mm diameter
radiation shield was used to reduce air temperature error caused by the sun except during the
following years: 2007, 2011-2014. During these years a separate data logger was used as the
primary source for air temperature and relative humidity observations. In 2013, our team
upgraded the station by replacing the wind sensors with two new units from Onset and a new
pyranometer from Apogee (http://www.apogeeinstruments.com/pyranometer) which exceeded
the 1277 Wm$^{-2}$ maximum reading of the previous Onset sensors (Covert, 2016). The
observations from LlanWX are largely continuous since 2007, with data gaps occurring
occasionally between 2010 and 2014.
Table 2.
This table details the variables, sensors, and their accuracy which are collected at the
meteorological stations throughout the region.

| Variable | Sensor | Accuracy | Unit |
|---|---|---|---|
| Air temperature | Onset HOBO S-THB-M002 Temperature RH Smart Sensor | ± 0.2 °C | °C |
| Precipitation | Onset HOBO S-RGB-M002 0.2 mm Rainfall tipping bucket Smart Sensor | ± 0.2 mm | mm |
| Relative humidity | Onset HOBO Temperature RH Smart Sensor: S-THB-M002 | ± 2.5 % | % |
| Wind speed | Onset HOBO S-WSB-M003 Wind Speed Smart Sensor | ± 1.1 m/s | m/s |
| Wind direction | Onset HOBO S-WDA-M003 Wind Direction Smart Sensor | ± 5 ° | ° |
| Incoming solar radiation | Onset HOBO S-LIN-M003 Solar Radiation Smart Sensor | ± 10 W/m² | W/m² |
| Atmospheric pressure | Onset HOBO S-BPB-CM50 Smart Barometric Pressure Sensor | ± 3.0 mb | mb |


A second AWS, situated at Portachuelo (referred to as PortWX in Figure 1), was installed
in July 2006 on a high pass at the top of the Llanganuco valley (4742 m a.s.l.). The station was
situated on a steep, rocky ridge between the Llanganuco valley and the Vaqueria valley. There
were only wind obstructions to the north due to a steep rock wall within 10 m of the station
location. The AWS at Portachuelo had the same sensors as its counterpart, LlanWX, however
was lacking an air temperature and relative humidity sensor with a radiation shield until it was
installed in July 2015. Prior to this upgrade, these variables were recorded each hour by a Lascar
data logger part of the ESN discussed in the following section. The Portachuelo station was
stolen in 2015 after recording data for nine years; a replacement station installed in 2016 was
also stolen so the site was abandoned. Gaps in weather station data is plotted in Figure A1 of the
Appendix.
Figure 3.
The typical setup for the HOBO automatic weather station, shown here is the LlanWX station
near the lower lake in Llanganuco. Measurements include air temperature, humidity, wind speed
and direction, incoming solar radiation, and rainfall. Note also the two Lascars with different
radiation shields hanging from crossbar. Photographs taken by R. Hellström (Covert, 2016).

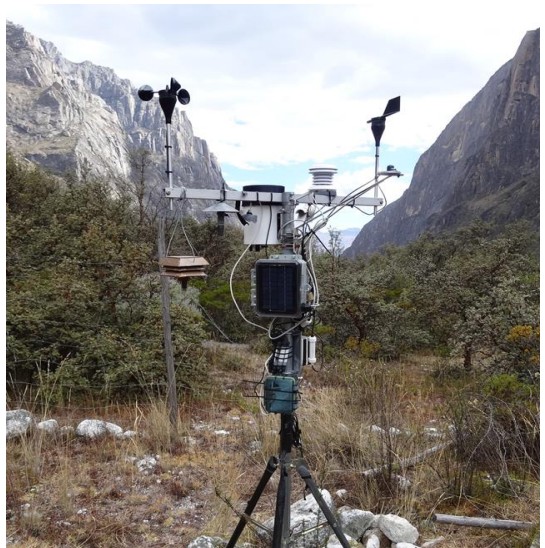


Smaller loggers measuring air temperature were installed in June 2005 to begin collecting
near surface temperatures to calculate lapse rates (Table 1). These data loggers were a series of 8
nickel sized iButton Thermochron® temperature loggers (Figure 3). Hellström et al. (2010)
demonstrated the effectiveness of using the Thermochron loggers for purposes of observing near
surface lapse rates within the Llanganuco valley between the elevations 3470 and 4740 m a.s.l.
(Covert, 2016). The ESN of iButtons was replaced in July 2006 with a more robust network of
Lascar El-USB2 data loggers (www.lascarelectronics.com) which were setup to measure air
temperature and relative humidity at one-hour intervals. Each Lascar logger is attached to a
custom-designed and locally crafted radiation shield made of a thin tin cone and two Styrofoam
pieces in order to reduce error caused by direct sunlight, as shown in Figure 3B. Figure 4 (from:
Covert, 2016) provides photos for a visual landscape context for each Lascar logger. Because the
LlanWX, LlanUp-4, and PortWX Lascar loggers are nearly entirely exposed to sunlight, the
recorded air temperature values in the dataset are higher than expected and contain greater error
than actual temperatures during the day (see section 4.1 for details). This Lascar network is still
recording hourly data. The Lascar network does not have any gaps in its time series; thus, it is
not included in the Appendix.
Figure 4.
Photos A-F show the locations of all Lascar dataloggers labeled by names matching Table 1.
Photographs taken by B. Mark and R. Hellström (Covert, 2016).

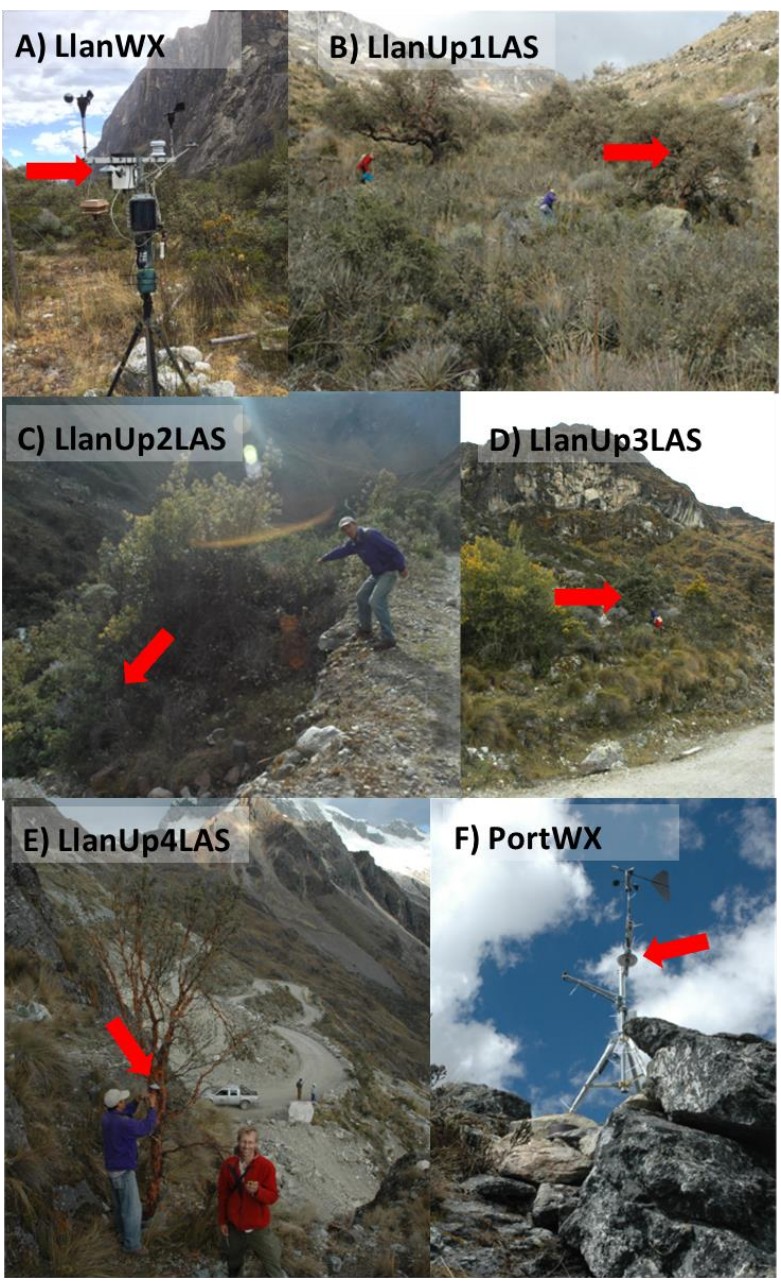


Quilcayhuanca contains two weather stations (Table 1) similar to the LlanWX station in
Llanganuco valley, the lower is located at 3924 m a.s.l. near an old river diversion station called
Casa de Agua (station referred to as CDAWX in Figure 1), and the upper is located at 4642 m
a.s.l. slightly above an alpine lake, Cuchillacocha (station referred to as CuchWX in Figure 1).
CDAWX collects all of the same variables as LlanWX (Table 2), is still active since its
installation in July 2013 and data are mostly continuous over this duration. CuchWX was also
installed in July 2013, is still presently active, and only has one data gap lasting longer than a
month (from November 2016 – June 2017). The data presented here conclude in 2020, but
ongoing data extensions will be provided through the repository.
**3.1.1 Relative Humidity Error Correction for LlanWX**
In 2006 and 2007 saturation of the humidity sensor at LlanWX resulted in relative
humidity values greater than 100% (as high as 110%) resulting from dust and condensation on
the humidity sensor and persisting during the wet season for periods of up to about 48 hours,
with periods of reasonable humidity values during drier periods. In light of data preservation,
these erroneously supersaturated values were simply replaced with 100% relative humidity; this
correction is referred to as "Lascar Error Adjustment" in Table 1. Note that the dew point was
equal to the air temperature for values of relative humidity 100% or greater, so no adjustment
was made for dew point. The humidity sensor was replaced in June 2007 and the correction was
no longer needed. Both the original and corrected values are retained in the LlanWX data file.
There was no similar humidity error observed for PortWX or any of the LAS sensors.
**3.1.2 Radiation Error Correction for Lascar Temperature Loggers**
A year-long comparison between the Lascar (Figure 5) and LlanWX AWS temperature
sensors indicated an expected error in the Lascar that was largely dependent on the time of day.
Variability in temperature bias due to solar radiative heating is likely caused by changes in solar
angle and cloud cover patterns, particularly during the dry season. It is well documented that
daytime temperature error rises on sunny days with calm wind and is reduced on windy and
cloudy days regardless of the season but is far more prominent during the dry season in the
tropical Andes (Georges and Kaser, 2002). It is important to note that snow cover, which can
create larger biases in temperature by reflected solar radiation, is minimal throughout the year at
the locations of the Lascars in this study. Over an entire month, nightly error was within the 0.5
°C resolution of the Lascar sensor. The occasional occurrence of greater nighttime error could be
caused by the sensor capturing longwave radiation emitted by the surface. The primary source of
consistent instrumentation error for air temperature measurements is the heating of radiation
shields from those locations exposed to direct sunlight, which required bias correction according
to Figure 5. We bias-corrected for radiation error using comparison with the LlanWX shielded
temperature and Lascar logger hanging next to it under the same exposure, including the
LlanWX, LlanUp-4 LAS, and PortWX station locations. We conducted this same comparison
when replacing Lascar sensors in 2015 and found no significant differences between old and new
sensors. Because solar radiation is the predominant source of error, corrections were only applied
to daylight hours between 07:00 to 19:00 (Covert, 2016). Future studies may consider thicker or
more encased radiation shields for temperature data loggers in tropical regions. Our group has
created and used a variety of styles of radiation shields as seen in Figure 5.
Figure 5.
Image A displays a Lascar data logger connected to its cone radiation shield. Two white
Styrofoam plates beneath the cone insulate the sensor from radiative heating. Image B shows the
radiation shields used on the AWS in comparison to the Lascar setup in image A. Image C shows
polylepis trees which are used to hide and shade Lascars in order to obtain accurate temperature
and relative humidity values. A comparison test between the Lascar setup and the AWS radiation
shield with no shading yielded the expected errors shown in graph D. Positive values indicate
that the Lascar reported a temperature higher than the AWS sensor. Photographs taken by R.
Hellström (Covert, 2016).

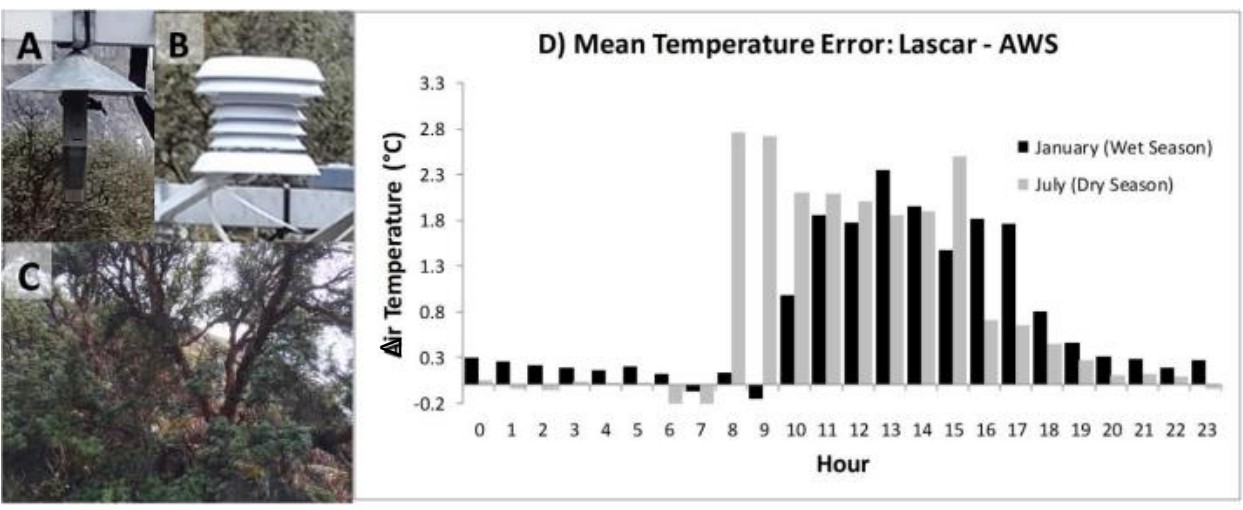


### 3.1.3 Data Gaps and Other Quality Control Checks

The most common reasons for gaps in weather data was either loss of sensors due to theft
or inability to access sensors because of logistical reasons, such as poor weather conditions or
human error during downloading or redeploying of sensors. In addition to corrections for solar
radiation heating and the relative humidity correction, humidity and temperature records were
assessed for accuracy and outliers removed if there were unrealistic deviations from the previous
24-hour trend of conditions within the valley. Most of these deviations occurred during time
intervals coincident with sensor deployment, battery replacement or data download, although
data (less than 1%) were also removed if there was a concern with the integrity of the sensor.
The four Lascar loggers were replaced or had new radiation shields installed in June 2015, and
likewise renamed in the dataset as LlanUp-1A, 2A, 3A, and 4A.

 **3.2 Hydrological Data**

The La Balsa gauge (8.87° S, 77.82° W at 1880 m a.s.l.) (Duke Energy, Orazul Energy,
Inkia Energy) near the diversion canal for the Cañon del Pato hydroelectric plant has a nearly
complete record since 1954, acting as a long-term reference point for discharge on the Rio Santa
at the base of the Callejon de Huaylas basin. Many other stream gauges and precipitation stations
were put in place by the hydroelectric companies, however, due to the lack of maintenance these
stations became unusable after the mid-1990's. In 2008, The Ohio State University, McGill
University, IRD, and the Peruvian glaciology unit of ANA commenced a joint project to
reinstitute a stream gauging network throughout the Cordillera Blanca (Baraer et al., 2012).
Many of these redeployed stream gauges in sub-catchments of the Rio Santa are still in working
condition and are continuing to be monitored.
The stream gauges comprise custom-designed and locally crafted steel stilling wells
containing two Solinst leveloggers (Figure 6), with one measuring total pressure (water plus
atmospheric), and the other measuring barometric pressure. The Model 3001 Solinst levelogger
Edge has an accuracy of ± 0.05 kPa and a resolution of 0.002% Full Scale (FS) for pressure and
a temperature accuracy of ± 0.05° C, and a temperature resolution of 0.003° C
(www.solinst.com). Water level is continuously monitored at these gauges at a temporal
resolution of 15-minutes by subtracting atmospheric pressure from the total pressure. The
adjusted water level variable in the discharge datasets indicates the adjustment based on where
the water level logger was located in the water column. This measurement is conducted in the
field using a built-in meter stick at each gauging station. Many factors influence the uncertainty
of discharge measurements, especially in smaller, turbid streams (McMillan et al., 2012). First,
the Solinst levelogger instruments used have accuracies and resolutions provided by the
company as a percentage of  FS as indicated above. Second, we estimate a ± 2 mm variation in
water stage due to the turbulent nature of the streams in the region. These streams have a variety
of bed surfaces, ranging from weedy to rocky, and flat to rolling, which are constantly modified
by high flows during each year. Each stage and discharge field measurement are influenced by
these small to large changes in the stream bed and the timing and location at which the discharge
is measured. Finally, stage-discharge rating curves inherently introduce variable amounts of error
depending on high or low flows, and how frequently they are updated. Stage-discharge rating
curves at all stations were established and verified by conducting discharge measurements using
the velocity area method and salt dilution during high flows (Figure A2 and Table A1 in the
Appendix). Prior to publishing this dataset, one updated rating curve was fitted for all the data at
each site, helping standardize the measurements and uncertainty in the data. All rating curves
developed from discharge measurements in the field are assessed by their fit and significance to
a quadratic function. All rating curves display $R^2$ values above 0.85, except for Cuchillacocha,
which provides an $R^2$ value of 0.71. This slightly lower $R^2$ value at Cuchillacocha is likely due to
the under-representation of high flows in the field measurements, and overall lower flow
volumes recorded at this site. Uncertainties (using standard error at 95% uncertainty intervals)
were calculated for all rating curves, indicating average error ranging from ± 3–20% for low
flows, and ± 4–70% during high flows. These error values are standard when using a stage-
discharge rating curve as described in Kiang et al. (2018), and McMillan et al. (2012). Rating
curves can be further constrained, and their variation better understood with the addition of more
high flow discharge point measurements (McMillan et al., 2010; Coxon et el., 2014).
Unavailable data is denoted by "NA" in the dataset. Linear interpolation of missing data was
only calculated for missing periods of less than 1 hour, coinciding with the time the leveloggers
were being downloaded in the field. Other than these short periods of interpolated data, the
datasets provided consist of raw data. Gaps of missing streamflow data are plotted in Figure A3
in the Appendix.
Figure 6.
Figure 6 illustrates the design of the stilling wells used to collect total pressure (water +
barometric) and barometric pressure. The photograph shows the stilling well and the insert tube
containing the pressure data loggers.

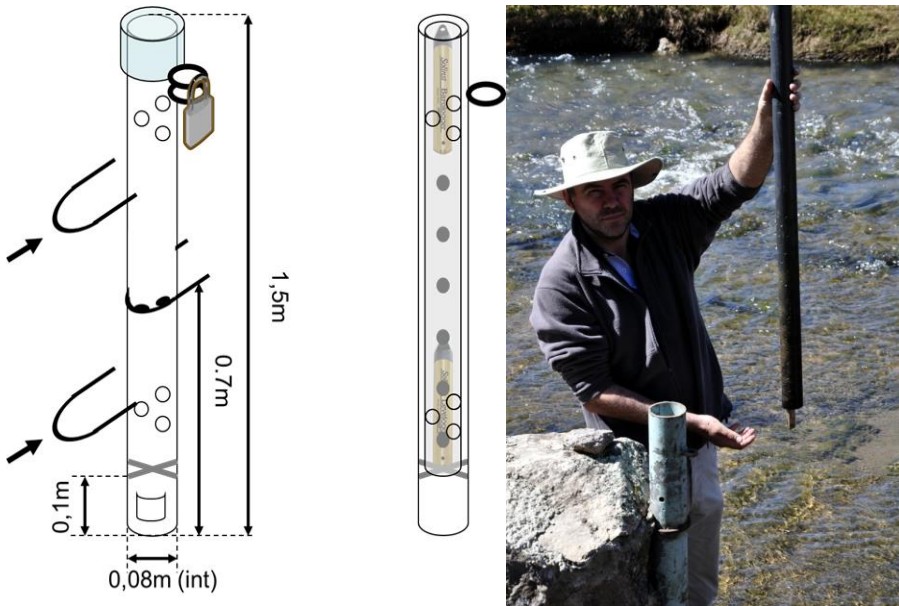


The stream gauges that were installed in 2008 and 2009 and which are continuing to
collect discharge are Casa de Agua, Cuchillacocha, Pachacoto, Llanganuco, and Querococha
(Figure 1). The Pumapampa gauge went out of commission in 2016 but is also included in this
dataset (Table 3). These stream gauges have variable lengths of record due to periods of missing
data, but still provide valuable streamflow information in a region where stream records are
limited. As shown in the map (Figure 1), two pairs of these stream gauges are located at different
points along the same Rio Santa tributary in their respective sub-catchments: Pumapampa and
Pachacoto in the Pachocoto valley; and Cuchillacocha and Casa de Agua in the Quilcayhuanca
valley. All discharge measurements are calculated from rating curves containing a minimum of
eight, point measurements throughout all times of the year. As mentioned above, rating curves
are not directly provided in this data paper because they are variable in nature as we collect new
discharge measurements in the field. The following two subsections provide a brief overview and
summary for each gauging station, organized by general location in the Cordillera Blanca,
moving from the southern end of the mountain range to the northern. Note that all glacier
coverage is calculated from RGI 6.0 (RGI Consortium, 2017).
Table 3.
This table provides geographical information about each of the hydrological gauging stations.
Elevations are recorded at bank of stream. Note that all glacier coverage is calculated from RGI
6.0 (RGI Consortium, 2017). Dates with an asterisk in the "Period of Operation" column indicate
the station is operational.

| Station | Station Valley | Latitude | Longitude | Elevation (m a.s.l.) | Period of Operation | Contributing Area (km²) | Percent basin covered in ice (%) |
|---|---|---|---|---|---|---|---|
| Cuchillacocha - CuchQ | Quilcayhuanca | -9.41 | -77.35 | 4631 | 2008-2019* | 4.1 | 60.49 |
| Casa de Agua - CDAQ | Quilcayhuanca | -9.46 | -77.37 | 3948 | 2009-2019* | 66.9 | 24.78 |
| Pumapampa - PumQ | Pachacoto | -9.88 | -77.24 | 4287 | 2008-2016 | 58.1 | 8.33 |
| Pachacoto - PachQ | Pachacoto | -9.85 | -77.4 | 3738 | 2008-2020* | 202.3 | 4.95 |
| Llanganuco - LlanQ | Ranrahirca | -9.07 | -77.65 | 3850 | 2008-2021* | 86.9 | 30.32 |
| Querococha - QuerQ | Querococha | -9.72 | -77.33 | 4005 | 2008-2019* | 62.9 | 1.53 |

## 3.2.1 The southern Cordillera Blanca
The Pumapampa catchment encompasses 58 km$^2$ with about 8% being covered by
glaciers (Table 3). Pumapampa gauge (referred to as PumQ in Figure 1) is located at 4287 m
a.s.l., below Nevado Pastoruri at the southern end of the Cordillera Blanca. This stream gauge is
situated in a channel through a low-lying meadow, where wet-season flow exceeds bankfull
stage, causing inaccuracies in high discharge measurements. Most notably, a series of values in
late-February 2014 indicated a peak discharge of 25 m$^3$s$^{-1}$, well above extremes measured from
the rating curve. It was determined that during this period the leveloggers were overrun by water
and measurements became unreliable solely during this flooding event. Outside of this event,
discharge at Pumapampa is consistently within the measured values on the rating curve. Mean
annual streamflow (averaged from three years of data missing less than one-month during the
entire year) at this stream gauge was 2.3 m$^3$s$^{-1}$ and a specific discharge of 1248 mm a$^{-1}$. Mean
streamflow during the dry season (May through September) in Pumapampa was 1 m$^3$s$^{-1}$, while
the mean streamflow during the wet season (October through April) was 2.9 m$^3$s$^{-1}$. Highest mean
monthly streamflow occurred in March 2014 with an average of 5.3 m$^3$s$^{-1}$ (Figure 7).
The Pachacoto gauge (referred to as PachQ in Figure 1) measures discharge 19 km
downstream from Pumapampa, at 3738 m a.s.l., just above its confluence with the Rio Santa.
Streamflow at Pachacoto is significantly greater than Pumapampa as it is the drainage point for a
larger area of 202 km$^2$, 5% of which is covered by glaciers. The mean annual streamflow at
Pachacoto was 3.2 m$^3$s$^{-1}$, averaged from the seven near-complete years of data, with a calculated
specific discharge of 499 mm a$^{-1}$. Dry season runoff accounted for approximately 20% of the
annual streamflow by volume, while the majority of the remaining wet season runoff occurred
during February and March.

Streamflow at another tributary catchment in the southern Cordillera Blanca is recorded
at the Querococha gauge (referred to as QuerQ in Figure 1) at 4005 m a.s.l., located 100 m
downstream of Querococha lake and defining a drainage area of 63 km$^2$, 1.5% of which is
glacierized. This drainage collects from two sub-catchments, one of which is entirely glacier-
free, while the other contains rapidly receding ice masses including the Yanamarey glacier.
Using five years of complete data, the mean annual streamflow at this gauging station was
calculated to be 1.4 m$^3$s$^{-1}$ and the specific discharge was 702 mm a$^{-1}$. The streamflow from this
gauge is somewhat regulated by Querococha, the second largest lake in the Cordillera Blanca.
This catchment has featured many previous research studies and many other instruments
throughout the basin are still consistently monitored by ANA.
Figure 7.
Figure 7 shows the monthly discharge, in m$^3$s$^{-1}$, at each of the gauging stations described in this
study calculated from all complete months of data for each station. The strong seasonal pattern is
clearly visible at most of the stream gauges.

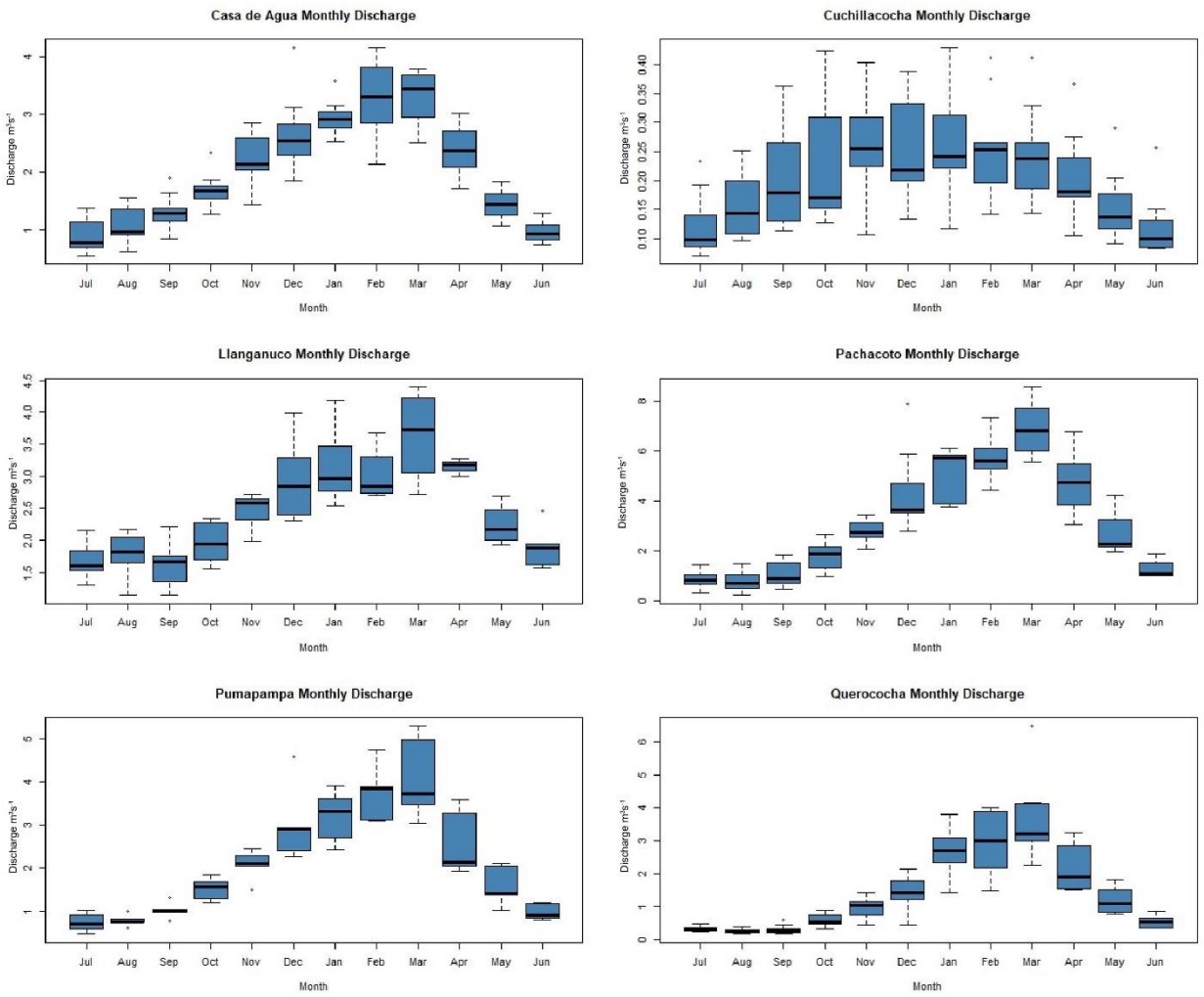


### 3.2.2 The central and northern Cordillera Blanca

Another catchment that contains two streamflow gauges is Quilcayhuanca, a valley
directly above the region's most populous city, Huaraz. The higher elevation stream gauge is
Cuchillacocha (referred to as CuchQ in Figure 1) which is situated at 4631 m a.s.l. and measures
discharge below a high-alpine lake and two cirque glaciers encompassing a drainage area of 4
km$^2$ (with 61% of the basin covered in ice). This station displays a noticeably different discharge
pattern throughout the year than the lower gauging stations which collect greater runoff.
Cuchillacocha discharge is not defined by a strong wet-dry season fluctuation, instead displaying
more variability each month and rising to peak values much earlier in the wet season than other
stream gauges. The average streamflow values at this location are also an order of magnitude
lower than other gauging stations due to the small drainage area it collects from (Figure 7).
The stream gauge located at a lower elevation in the Quilcayhuanca valley is Casa de
Agua (referred to as CDAQ in Figure 1) at 3948 m a.s.l. This gauging station is found near a
channel cut in the stream, in a large meadow, where water was previously rerouted for
agricultural purposes. This gauge collects drainage from an area of 67 km$^2$ below the confluence
of two upper catchments, Cuchillacocha and Cayesh, well above the city of Huaraz (3100 m
a.s.l.). From this drainage point in the watershed, the basin is approximately 25% glacierized.
The mean annual streamflow, calculated from six complete years of data at Casa de Agua, was 2
m$^3$s$^{-1}$, with a specific discharge of 943 mm a$^{-1}$.
The last streamflow gauge still recording data, Llanganuco (referred to as LlanQ in
Figure 1), is named after the valley it is located in and is positioned at 3850 m a.s.l., below two
valley lakes and a catchment area of 87 km$^2$, 30% of which is currently glacierized. The mean
annual streamflow, calculated from three complete years of data, was 2.3 m$^3$s$^{-1}$, with a specific
discharge of 835 mm a$^{-1}$. A stream gauge has been in commission at Llanganuco on and off for
68 years, beginning as a gauge for UGRH and hydroelectric companies and later, after the
original station became defunct, a site for our new, higher temporal resolution network of
gauges.

## 4 Data availability

The datasets presented here are available freely from
https://doi.org/10.4211/hs.35a670e6c5824ff89b3b74fe45ca90e0 (Mateo et al., 2021). The
hydrological and meteorological data from the Cordillera Blanca have been used in short
segments in previous studies (Baraer et al., 2012; Baraer et al., 2015). Data availability varies
from station to station depending on location of loggers and instrumental errors which caused
periods of time to lapse without data being recorded. These datasets represent a majority of the
streamflow and meteorological data collected by our research group over the past two decades
collected at a high temporal resolution of 15-30 minutes. These include point discharge
measurements and short-time periods of 1-minute temporal resolution meteorological
measurements. The data provided here concludes in 2020, however future data collected by these
stations will be added to an updated version of the repository as future field seasons occur and
the effort to provide hydrometeorological data to the scientific community will continue at all of
the involved universities and institutions.

## 5 Conclusions

The Cordillera Blanca in the tropical Andes of Peru is a unique, high mountain region
where high resolution meteorological and hydrological time series observations collected from a
network of instruments over the past two decades have been compiled in a new dataset. The
region has been the focus of glacier monitoring efforts for nearly a century. While daily-to-
monthly time series of meteorological and hydrological observations have been recorded
discontinuously for nearly 70 years, there was no generalized sub-daily data monitoring until the
early 2000s. Maintaining this network of instruments recording high-temporal resolution data has
involved traveling to remote areas of the Cordillera Blanca, protecting the instruments from the
harsh weather conditions at high elevations, and concealing instruments to prevent theft. There is
also difficulty in developing long-term strategies because most funding agencies are focused on
short-term based projects and do not fund continuous monitoring and maintenance of data
collection networks. In the context of ongoing global climate change with variable localized
extremes, long-term monitoring of hydrometeorological variables is of utmost importance to
document and understand the changes that are occurring.
The datasets collected and described in this paper support investigations into climate
evolution, water resource availability, and hydrological changes across the Cordillera Blanca in
the past twenty-years. These datasets will also provide valuable, easy-to-access observations for
local water resource managers. This paper provides an overview of the variables which have
been measured and what is being made available as of 2021. Future measurements recorded in
the field will be made available as they are collected, to further build on the available hydro-
meteorological database in the Cordillera Blanca.














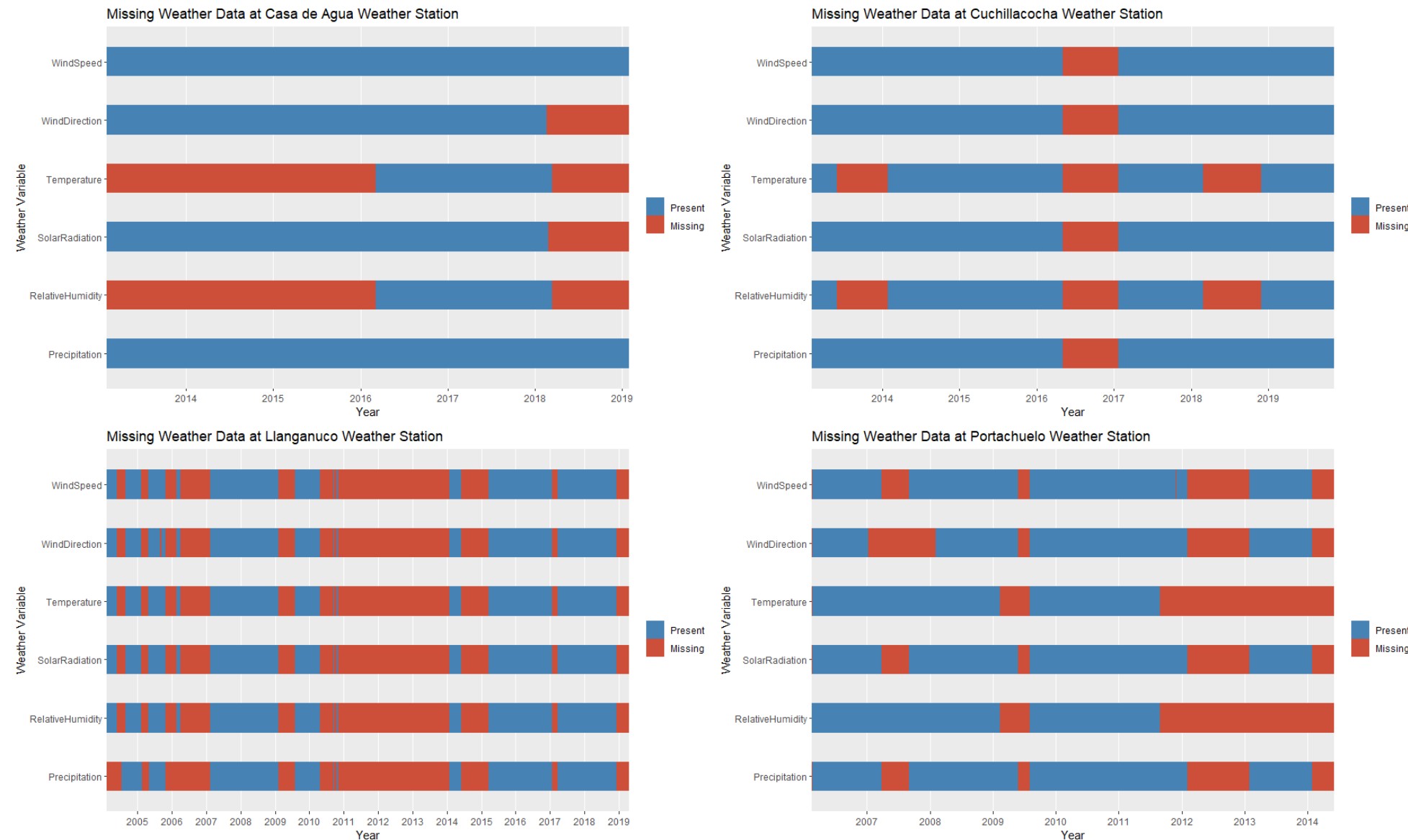

Figure A1. Plot of data gaps in the weather station data at each weather station for all variables collected, available at https://doi.org/10.4211/hs.35a670e6c5824ff89b3b74fe45ca90e0.

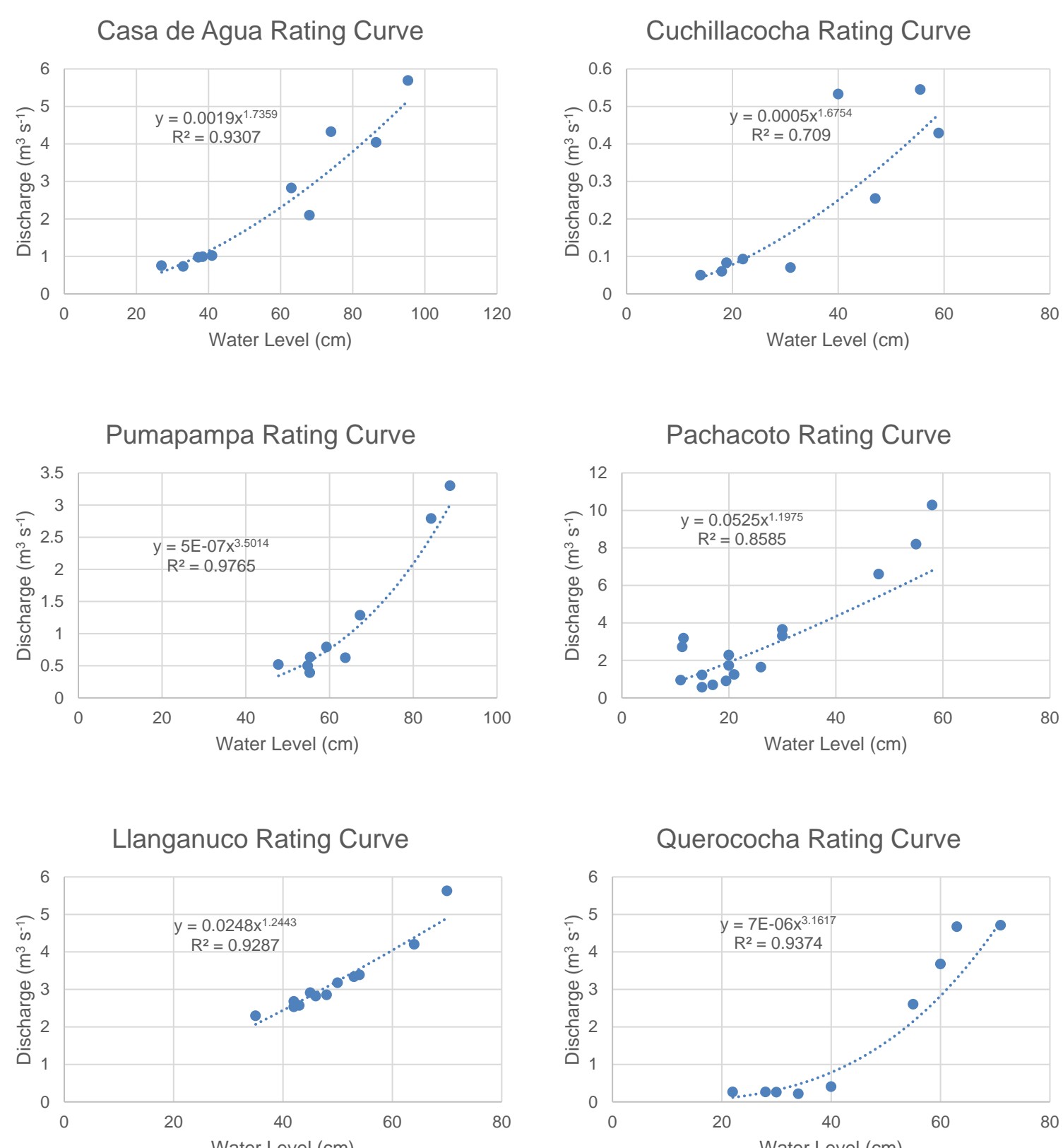

Figure A2. Rating curves calculated for each discharge gauging station

Table A1. Rating curve values used for each discharge gauging station

| Casa de Agua | | | Cuchillacocha | | |
|---|---|---|---|---|---|
| Date | Water level (cm) | Discharge (m$^3$ s$^{-1}$) | Date | Water level (cm) | Discharge (m$^3$ s$^{-1}$) |
| 7/6/08 13:40 | 41 | 1.019999981 | 2/3/09 10:20 | 59 | 0.42899999 |
| 11/5/08 13:30 | 68 | 2.095999956 | 4/27/09 16:50 | 47 | 0.254299998 |
| 2/2/09 10:20 | 86.5 | 4.043000221 | 11/18/09 9:45 | 55.5 | 0.544799984 |
| 4/27/09 12:15 | 63 | 2.825000048 | 3/18/13 15:00 | 40 | 0.533 |
| 3/19/13 13:00 | 95.33 | 5.689 | 7/6/08 14:40 | 31 | 0.07 |
| 7/17/13 8:30 | 27 | 0.751999974 | 7/17/13 12:30 | 22 | 0.092600003 |
| 2/9/14 17:15 | 74 | 4.325 | 6/22/17 15:45 | 18.9 | 0.083194 |
| 6/23/17 11:28 | 38.4 | 0.99 | 7/6/18 13:00 | 14 | 0.05 |
| 7/5/18 11:51 | 33 | 0.73 | 7/3/19 14:50 | 18 | 0.06 |
| 7/4/19 10:45 | 37.2 | 0.977 | | | |

| Pumapampa | | | Pachacoto | | |
|---|---|---|---|---|---|
| Date | Water level (cm) | Discharge (m$^3$ s$^{-1}$) | Date | Water level (cm) | Discharge (m$^3$ s$^{-1}$) |
| 2012 | 47.795 | 0.518 | 7/4/2005 | 15 | 1.224 |
| 2012 | 55.295 | 0.391 | 7/4/2005 | 19.5 | 0.9 |
| 2012 | 59.295 | 0.791 | 7/4/2005 | 20 | 2.285 |
| 2012 | 55.395 | 0.6345 | 7/4/2005 | 26 | 1.64 |
| 2012 | 67.295 | 1.285 | 7/4/2005 | 30 | 3.653 |
| 2012 | 84.295 | 2.79 | 4/21/2010 | 30 | 3.309 |
| 7/1/2013 | 54.795 | 0.499037569 | 7/1/2013 | 11 | 0.9424 |
| 6/1/2014 | 63.795 | 0.6229 | 7/6/2005 | 17 | 0.6915 |
| 2/1/2014 | 88.795 | 3.3 | 6/23/13 10:30 | 11.525 | 3.179 |
| | | | 6/23/13 11:30 | 11.305 | 2.715 |
| | | | 7/6/2005 | 55 | 8.197 |
| | | | 3/25/14 13:45 | 58 | 10.287 |
| | | | 6/1/2014 | 20 | 1.727 |
| | | | 7/6/2005 | 48 | 6.6 |
| | | | 6/24/2017 | 21 | 1.244 |
| | | | 7/8/2018 | 15 | 0.563 |

| Llanganuco | | | Querococha | | |
|---|---|---|---|---|---|
| Date | Water level (cm) | Discharge (m$^3$ s$^{-1}$) | Date | Water level (cm) | Discharge (m$^3$ s$^{-1}$) |
| 4/30/03 7:30 | 50 | 3.177 | 9/16/08 15:20 | 22 | 0.264400005 |
| 9/5/03 10:30 | 45 | 2.911 | 12/3/09 9:35 | 60 | 3.677999973 |
| 6/30/03 7:12 | 43 | 2.57 | 7/13/10 10:33 | 28 | 0.266099989 |
| 7/31/03 9:00 | 42 | 2.68 | 7/13/13 15:55 | 30 | 0.2588 |
| 9/3/03 7:00 | 46 | 2.82 | 3/26/14 10:00 | 71 | 4.711 |
| 10/15/03 9:50 | 54 | 3.387 | 2/9/14 13:15 | 55 | 2.603 |
| 12/3/03 11:00 | 64 | 4.202694736 | 2/19/14 11:00 | 63 | 4.673 |
| 5/5/05 9:30 | 53 | 3.336 | 6/27/17 12:57 | 40 | 0.406 |
| 6/29/05 10:45 | 35 | 2.295 | 7/8/18 17:41 | 34 | 0.217 |
| 10/7/05 15:30 | 42 | 2.534 | | | |
| 10/31/05 16:15 | 48 | 2.853 | | | |
| 3/24/14 13:00 | 70 | 5.627 | | | |

Figure A3. Plot of data gaps in the discharge data, organized by gauging station, available at

https://doi.org/10.4211/hs.35a670e6c5824ff89b3b74fe45ca90e0.

604

**Author Contribution**

EM prepared the manuscript with contributions from co-authors. EM, RH and MB curated the
data in preparation for this article. BGM acquired the initial funding for this international
research collaboration and instrumental network. All authors collaborated in fieldwork gathering
data and maintaining the instruments and dataloggers.

**Competing Interests**

The authors declare that they have no conflict of interest.

**Acknowledgements**

Funding for instrumentation, installation and field maintenance was provided by various
agencies and institutions associated with overlapping research projects and educational initiatives
through respective institutions. BGM acknowledges Ohio State University, National Science
Foundation (NSF #0752175 with REU supplement; NSF #1010384; NSF #1316432), National
Geographic Society, NASA (New Investigator Program), and US Fulbright (Award #4506).

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
