# Peer review of "High temporal resolution hydrometeorological data collected in the tropical Cordillera Blanca, Peru (2004-2020) Emilio I. Mateo1, Bryan G. Mark1, Robert Å. Hellström2, Michel Baraer3, Jeffrey M. McKenzie4, Thomas Condom5, Alejo Cochachín Rapre6, Gilber Gonzales6, Joe Quijano Gómez6, Rolando Cesai Crúz Encarnación6 1Department of Geography, Byrd Polar and Climate Research Center, The Ohio State Uni"

_Earth System Science Data, 2021_

## Author Response (AR1)

ESSD Responses

**Anonymous Referee #1**

**RESPONSE:**

Thank you for your comment on our discussion paper. We appreciate all of your feedback and address your comments and suggestions in the following response. While we agree that it is unlikely that the people responsible for theft of the equipment would find that information on ESSD, providing that information publicly makes it relatively easy to find online. We are providing rough coordinates for the discharge stations, and we are happy to provide exact locations of weather stations upon request. The map has been updated to show bold lines for each stream to make them more visible to the viewer. We have taken your suggestions of editing the meteorological station names in the data and have removed rating curves from the data, these will now be available upon request. Finally, the uncertainties you are addressing and which are described in this paper are related to the Lascar meteorological data, not the discharge data. The raw pressure values for both water and barometric pressures are provided in the discharge data files. We have added a note about the adjusted water level variable found in the discharge dataset. Thank you again for your comments.

**Anonymous Referee #2**

**RESPONSE:**

Thank you for your comment on our discussion paper. We appreciate your feedback and address your general comments and line-by-line suggestions in the following response.

General comment response: We are now including a section to address the data processing and quality control that was performed on the data prior to writing this paper. Many of the quality checks, as you noted, were related to the weather or Lascar logger data, while the discharge data was found to be reliable at the onset. We have addressed the concern of quality control on the weather data and the impacts of changes in instrumentation by including new subsections of section 3. We also include a citation and link to SENAHMI, however this data is not included in our article because this is not data we have collected. We have also added to the tables already included in the paper to address the temporal resolution of the data, and the period of operation for each logger. We have also decided not to include the rating curves in this paper because they are variable by nature, and will be updated with each visit to the field sites. We are happy to provide these to readers upon request. We appreciate the interest in some of the summary statistics for the data, however, as this is a data paper, we are not providing that information in the paper itself. We also are glad that you are interested in further assessments of this data, and additional analyses will be forthcoming based on current ongoing research using this data.

We have gone back through the datasets themselves and made all time information and header information uniform for clarity. We have also removed special characters. As mentioned in our other comment response, we have provided coarse geographic coordinates for the discharge stations, however, we have refrained from providing precise locations (apart from the visual location on the map) for the weather stations because of the high-risk of theft in this region.

We have made corrections based on your line by line comments. Thank you for these helpful suggestions; we have provided more information in certain locations. As far as we are aware of, the SENAHMI precipitation data are of daily temporal resolution. This data was also not collected by us, and thus was not included in this paper. We have added a clarifying sentence for Lascar error adjustment in a following paragraph detailing information about the Lascars. We appreciate your comment on Figure 4, but have retained the images to provide a visual landscape context for the Lascar loggers; not only for readers who are less familiar with the Cordillera Blanca, but in case additional location-specific issues arise as users scrutinize the data. We have also added m3/s in the caption for Figure 6 for clarification. We have provided all information necessary to calculate specific discharge and would encourage readers interested in further analyses to use this information. The updated Lascar sensors were compared to their previous counterparts to ensure data quality continuity, which is also now addressed in section 3.1.2.

Thank you again for your detailed comments and suggestions. We feel that the paper is greatly improved because of the comments you provided.

---

## Editor Decision (ED1)

**Editor comments – minor revision** 04.04.2022

Thank you for the revision of the manuscript and the improvement on the repository. However, I still have some issues with some issues in the manuscript, the supplement and the Readme file. Please consider my comments and revise the documents once more.

1) Manuscript:

- Data basis / General comment:
  o The basis for an ESSD paper is a published dataset on a trusted repository with a fixed DOI. You cannot refer to a dataset that is still going on in the paper, as you also cannot describe the features/uncertainties of this dataset appropriately if it has not been collected yet. If your data collection is still going on, the best option is to refer to a fixed dataset in the past for the paper, but already mention in the paper that data collection is continuing and that you will update the dataset on the repository regularly, leading to new versions with extended time series. The paper will, however, always refer to a fixed dataset version with a fixed DOI.
    ➔ Please make your time period/fixed dataset which you are referring to with the paper clear in the manuscript and mention the possible extensions that will happen on the repository, if you plan to do so.

- Rating curves:
  o Thank you for the extension of the discussion on the uncertainties of the discharge measurements. This is already very helpful. However, it would still be good to include the rating curves (and their support/uncertainties) which you finally used as a basis for the stage-discharge relationship of the data in the paper with its fixed DOI. Of course this may change once you continue measurements, but as a snapshot and a basis for assessing which ranges of the data are more trustworthy than others the curves are very valuable, especially if it is not completely clear at which values the "low flows" or "high flows" with their considerably different error range can be expected. Updated rating curves could then still be provided on request, but for the published dataset the rating curve should be definite (and included in an appendix for example), not fluid.

2) Supplement:

- Displaying the times when sensors are working and when they are not in a figure is a good way. However, the presentation of it is possibly not the most intuitive/easies to grasp. Some suggestions:
  o Please refer to the dataset that is actually in the paper, with its fixed DOI.
  o yearly resolution seems a bit coarse, maybe provide further ticks or lines to make comparisons of the durations a bit easier, possibly on a monthly basis?
  o The rows are very wide with no apparent reason for it. If you made them narrower, you could possibly group more corresponding together, for example the weather

station and discharge data from the same catchment. A raster with small monthly blocks that are black and white or something might also work for a compressed display.
- The information of availability for the LASCAR data is missing. Please also add it.
- Please be consistent in the use of abbreviations for the station names. For example, in the map Fig. 1 and Table 3 you write the discharge stations as CuchQ, CDAQ, PumQ etc. whereas in the supplement they are called cuchilla, cda, puma etc. It just makes it easier for the reader to know that the same thing is meant.
- Please add this information as an appendix rather than a supplement (or if the compressed graph is compact enough, it could even be a figure), it is an important overview of the data, especially in a data-sparse region such as Peru, so it would be good to have it attached directly to the paper.

3) Readme file:

- File format:
  o .xlsx is a proprietary format. Please put it in as a PDF so that everybody can read it
  o A form of text file (instead of this table format) is easier to read, insert the tables and maps into a text file rather than having text blocks in a table.
- Tables: In the form it is now, last column doesn't match first ones as the lines of the last column don't correspond to the lines in the other columns. Please put this information separately.
- Last header doesn't really talk about the sensors, the sampling and the data, rather about the cooperation which is more like an acknowledgement bit than needed to work with the data. More helpful would be to actually name the sensors that are measuring (or put them into the tables), the accuracy and also the overview of the time series when everything was measuring (like in the supplement/appendix).
- You don't need to duplicate the abstract which is on the repository once more in the readme. However, providing information on sampling/sensors, and data processing for the individual variables is relevant and necessary to be able to use the data.
- Maybe some pointers about the documentation alongside the data (as in the readme) can be found here (under "Documentation and metadata"). Not everything may apply, but it provides a good list of what should be found on the repository, either on the landing page (eg. authors etc. are already there) or in the readme.
  o https://dataservices.gfz-potsdam.de/portal/drr.html

---

## Author Response (AR2)

**ESSD Point-by-Point Responses to Referees**

**Referee #1**

In response to the comment from Referee #1 about our suggestions for future setups of temperature sensors, we have added two short sentences and a reference to Figure 5 where we show the types of radiation shields our team has used and developed over the study period.

The comment relating to Table 3 has been addressed in the caption for the table. These elevations are recorded at the bank of the stream, thus are valid as reference points.

**Referee #2**

In response to the comment (1) from Referee #2 about the overview of the data, we have added the necessary information that was missing. Much of the requested information is already found in Table 1 and 3, and the text. This includes:

- data length (Table 1 and 3),
- descriptions, units, and sampling rates (resolution) (found in Tables 1, 2, and 3, and the text)
- abbreviations beyond what is included are now unnecessary because these all align with the newly updated datasets

We have added sentences to more clearly identify gaps in discharge data, while information related to the missing weather station data was already included in the text. The only other addition that was requested here was the type of sampling, which we added in the text (ln. 262) by saying point measurements, indicating a sample point. The description of the discharge data collection (ln. 401-415) clearly indicates the sample point measurements of the discharge data. We feel that these edits succinctly address your comment (1) regarding the dataset descriptions, and we feel that the tables we have included provide a significant amount of valuable information in conjunction with the body of text.

In response to comment (2), the data had all been modified, but not updated/published in the hydroshare system, they are now available in their final form. The naming is now consistent across all datasets.

In response to comment (3), Figure 7 (incorrectly labeled by referee as 6) has been replotted, "Discharge $m^3s^{-1}$" now appears as the y-axis label for each station in the figure.

---

## Author Response (AR3)

**Author Responses to Editor Comments**

**1**) The data on the repository: While the ESSD paper provides the opportunity to describe the data, measurements and field sites in much detail, the information on the repository should be sufficient alone to work with the data without the manuscript and needs some improvement.

a. First of all, the description is simply a copy of the paper abstract. This is confusing because some of the wording does not fit. ("this article", "this paper")

This has now been addressed, and does not refer to the dataset as "this paper" or "article."

b. While the general abstract is suited to give an overview of the intentions behind the measurements, there should be a readme file alongside the data,

(i) Listing the sites and the respective measurements, probably also including the map from the paper.

(ii) A brief description of sampling methods and data processing.

(iii) Describing the folder structure including the naming of the files and also of the columns within the files.

A readme file has been created and added to the data repository. All information requested above is now included in this readme file.

2) The overview table that Referee #2 requested: I also think this is a very valuable asset for the users of the dataset, to have the information which of the measurements is available for which time periods.

The sentences you added for the streamflow data just indicate that there are gaps. For modellers for example, the information when there are meteorological and streamflow data in the same area available so that they can judge the length of consistent time series for the models would be incredibly valuable. A suggestion would be to structure the table additionally in a way so that it is visible which of the datasets belong to the same catchment, without looking up the station names.

We have decided that a figure showing all gaps in the datasets would be more effective than a lengthy table. We have added these figures as a supplement so it can be found as needed by the user. These figures are also referenced in the paper in appropriate locations.

Also, please indicate which is the period you are referring to within this paper and the referenced dataset. If stations are still running you can indicate that with a symbol or something, but the basis for the manuscript is a definite dataset.

The period of operation for all stations is listed in Table 1 and 3. Their operational status has been updated in the table with an asterisk as an indicator and has been reflected upon in the caption for both tables.

The current status of data acquisition is addressed in lines 226-229:

The supplemental figures S1 and S2 indicate missing data beyond 2020, however due to travel restrictions over the past two years we have not been able to collect the up-to-date measurements. Our intent is to sustain these hydrological and meteorological measurements into the future once travel to the region is possible again.

**3**) Uncertainty/errors of discharge measurements: Although Referee #1 did not insist on the rating curves, a measure of uncertainty and a discussion about this is still needed in the data and manuscript.

Did you use the rating curve for all the measurements in the dataset? Did the rating curves change? Based on the quality of the rating curve and the uncertainty in the water level measurements you can give an uncertainty range for the discharge measurements. If the curves changed and you did not update older data, then the discharge values would have a different basis for the uncertainty range as well. So please add this information.

Uncertainty in discharge measurements is not directly calculated in this paper for a variety of reasons. We have however added some statements in the paper to address the possible uncertainties and have added some discussion of the range of uncertainty in the rating curves. We have also added citations that support our stance of providing R2 and error percentages for rating curves, while also describing the fluid nature of the stream sites. Our citations also provide evidence that the errors we are seeing in rating curves is typical for hydrological data when using stage-discharge rating curves.

This has now been fully addressed in Lines 417-440:

Many factors influence the uncertainty of discharge measurements, especially in smaller, turbid streams (McMillan et al., 2012). First, the Solinst levelogger instruments used have accuracies and resolutions provided by the company as a percentage of FS as indicated above. Second, we estimate  $a \pm 2$  mm variation in water stage due to the turbulent nature of the streams in the region. These streams have a variety of bed surfaces, ranging from weedy to rocky, and flat to rolling, which are constantly modified by high flows during each year. Each stage and discharge field measurement are influenced by these small to large changes in the stream bed and the timing and location at which the discharge is measured. Finally, stage-discharge rating curves inherently introduce variable amounts of error depending on high or low flows, and how frequently they are updated. Stage-discharge rating curves at all stations were established and verified by conducting discharge measurements using the velocity area method. Prior to publishing this dataset, one updated rating curve was fitted for all of the data at each site, helping standardize the measurements and uncertainty in the data. All rating curves developed from discharge measurements in the field are assessed by their fit and significance to a quadratic function. All rating curves display R2 values above 0.85, except for Cuchillacocha, which provides an R2 value of 0.71. This slightly lower R2

value at Cuchillacocha is likely due to the under-representation of high flows in the field measurements, and overall lower flow volumes recorded at this site. Uncertainties (using standard error at 95% uncertainty intervals) were calculated for all rating curves, indicating average error ranging from  $\pm 3$ –20% for low flows, and  $\pm 4$ –70% during high flows. These error values are standard when using a stage-discharge rating curve as described in Kiang et al. (2018), and McMillan et al. (2012). Rating curves can be further constrained, and their variation better understood with the addition of more high flow discharge point measurements (McMillan et al., 2010; Coxon et el., 2014). All rating curves are available upon request.

If you used one rating curve for and backcorrected all water level values, it would be also good to put the curves for the different stations as graphs in the appendix and the data of the rating curve measurements in the repository.

The other authors and I have again discussed the request for rating curves to be included in the paper as a supplement. We would greatly prefer to leave the rating curves as available upon request because of the fluid nature of these equations. As indicated in the newly added text, each station has one rating curve, and is "backcorrected" to all water level values in each time series and is fully up to date with all recent data in each time series.

---

## Author Response (AR5)

**Author Responses to Editor Comments**

1) Manuscript:

- Data basis / General comment:
o The basis for an ESSD paper is a published dataset on a trusted repository with a fixed DOI. You cannot refer to a dataset that is still going on in the paper, as you also cannot describe the features/uncertainties of this dataset appropriately if it has not been collected yet. If your data collection is still going on, the best option is to refer to a fixed dataset in the past for the paper, but already mention in the paper that data collection is continuing and that you will update the dataset on the repository regularly, leading to new versions with extended time series. The paper will, however, always refer to a fixed dataset version with a fixed DOI.
☐ Please make your time period/fixed dataset which you are referring to with the paper clear in the manuscript and mention the possible extensions that will happen on the repository, if you plan to do so.

This has been updated throughout the paper and is clearly referring to the fixed dataset, with one or two references to updates in the future so the reader knows that future data will be added to a repository related to the one published here.

- Rating curves:
o Thank you for the extension of the discussion on the uncertainties of the discharge measurements. This is already very helpful. However, it would still be good to include the rating curves (and their support/uncertainties) which you finally used as a basis for the stage-discharge relationship of the data in the paper with its fixed DOI. Of course this may change once you continue measurements, but as a snapshot and a basis for assessing which ranges of the data are more trustworthy than others the curves are very valuable, especially if it is not completely clear at which values the "low flows" or "high flows" with their considerably different error range can be expected. Updated rating curves could then still be provided on request, but for the published dataset the rating curve should be definite (and included in an appendix for example), not fluid.

Rating curves have been added as an appendix.

2) Supplement

- Displaying the times when sensors are working and when they are not in a figure is a good way. However, the presentation of it is possibly not the most intuitive/easiest to grasp. Some suggestions:
o Please refer to the dataset that is actually in the paper, with its fixed DOI.

This has now been listed in the captions.

o yearly resolution seems a bit coarse, maybe provide further ticks or lines to make comparisons of the durations a bit easier, possibly on a monthly basis?
o The rows are very wide with no apparent reason for it. If you made them narrower, you could possibly group more corresponding together, for example the weather station and discharge data from the same catchment. A raster with small monthly blocks that are black and white or something might also work for a compressed display.

Monthly was attempted but is very cramped and unreadable in the format described here. New plots were created, and space was given in between variables to make them more legible. The style is adapted from previous ESSD data papers (Strasser et al., 2018; https://doi.org/10.5194/essd-10-151-2018) which have been successful in displaying missing data.

- The information of availability for the LASCAR data is missing. Please also add it.

Lascar loggers are not missing data, which is why they are not included. Please look at the data. A sentence has been added to the manuscript to convey this.

- Please be consistent in the use of abbreviations for the station names. For example, in the map Fig. 1 and Table 3 you write the discharge stations as CuchQ, CDAQ, PumQ etc. whereas in the supplement they are called cuchilla, cda, puma etc. It just makes it easier for the reader to know that the same thing is meant.

All abbreviations have been updated in these figures.

- Please add this information as an appendix rather than a supplement (or if the compressed graph is compact enough, it could even be a figure), it is an important overview of the data, especially in a data-sparse region such as Peru, so it would be good to have it attached directly to the paper.

All of the above mentioned figures, along with the rating curves have been added to an appendix, and the supplement has now been removed entirely because of this shift.

3) Readme file:

- File format:
o .xlsx is a proprietary format. Please put it in as a PDF so that everybody can read it
o A form of text file (instead of this table format) is easier to read, insert the tables and maps into a text file rather than having text blocks in a table.

This has been updated.

- Tables: In the form it is now, last column doesn't match first ones as the lines of the last column don't correspond to the lines in the other columns. Please put this information separately.

This has been edited appropriately.

- Last header doesn't really talk about the sensors, the sampling and the data, rather about the cooperation which is more like an acknowledgement bit than needed to work with the data. More helpful would be to actually name the sensors that are measuring (or put them into the tables), the accuracy and also the overview of the time series when everything was measuring (like in the supplement/appendix).

The last header has been removed and the entire readme file has been revamped to include all information addressed here.

- You don't need to duplicate the abstract which is on the repository once more in the readme. However, providing information on sampling/sensors, and data processing for the individual variables is relevant and necessary to be able to use the data.

Abstract is removed from the readme file.

- Maybe some pointers about the documentation alongside the data (as in the readme) can be found here (under "Documentation and metadata"). Not everything may apply, but it provides a good list of what should be found on the repository, either on the landing page (eg. authors etc. are already there) or in the readme.

Bullet points for each form of data and sensor has been added at the beginning of the readme file. Tables and figures have been included and are numbered for ease of corresponding to the written portion of the readme file.